# Benzocarbazoledinones as SARS-CoV-2 Replication Inhibitors: Synthesis, Cell-Based Studies, Enzyme Inhibition, Molecular Modeling, and Pharmacokinetics Insights

**DOI:** 10.3390/v16111768

**Published:** 2024-11-13

**Authors:** Luana G. de Souza, Eduarda A. Penna, Alice S. Rosa, Juliana C. da Silva, Edgar Schaeffer, Juliana V. Guimarães, Dennis M. de Paiva, Vinicius C. de Souza, Vivian Neuza S. Ferreira, Daniel D. C. Souza, Sylvia Roxo, Giovanna B. Conceição, Larissa E. C. Constant, Giovanna B. Frenzel, Matheus J. N. Landim, Maria Luiza P. Baltazar, Celimar Cinézia Silva, Ana Laura Macedo Brand, Julia Santos Nunes, Tadeu L. Montagnoli, Gisele Zapata-Sudo, Marina Amaral Alves, Diego Allonso, Priscila V. Z. Capriles Goliatt, Milene D. Miranda, Alcides J. M. da Silva

**Affiliations:** 1Instituto de Pesquisa de Produtos Naturais, Universidade Federal do Rio de Janeiro, Ilha do Fundão, CCS, Bloco H—Sala H29, Rio de Janeiro 21941-902, RJ, Brazil; luanagoncalvessouza10@gmail.com (L.G.d.S.); silva.cordeiroj@gmail.com (J.C.d.S.); edgar.ippn@gmail.com (E.S.); ju.villar.guimaraes@gmail.com (J.V.G.); dennis.maia.dm@gmail.com (D.M.d.P.); marina.amaral@ippn.ufrj.br (M.A.A.); 2Programa de Pós-Graduação em Modelagem Computacional, Grupo de Modelagem Computacional Aplicada, Universidade Federal de Juiz de Fora, Juiz de Fora 36036-900, MG, Brazil; dudaalves.penna@gmail.com (E.A.P.); carius.nara@gmail.com (V.C.d.S.); mjnlandim76@gmail.com (M.J.N.L.); baltazar.maria@estudante.ufjf.br (M.L.P.B.); 3Laboratório de Morfologia e Morfogênese Viral, Instituto Oswaldo Cruz, Rio de Janeiro 21041-250, RJ, Brazil; alicerosa@aluno.fiocruz.br (A.S.R.); vivian.ferreira@ioc.fiocruz.br (V.N.S.F.); danielsouza@aluno.fiocruz.br (D.D.C.S.); sylvia.roxo@ioc.fiocruz.br (S.R.); giovannaconceicao@id.uff.br (G.B.C.); 4Programa de Pós-Graduação em Biologia Celular e Molecular, Instituto Oswaldo Cruz, Rio de Janeiro 21041-250, RJ, Brazil; 5Laboratório de Biotecnologia e Bioengenharia Tecidual, Instituto de Biofísica Carlos Chagas Filho, Universidade Federal do Rio de Janeiro, Ilha do Fundão, CCS, Rio de Janeiro 21941-902, RJ, Brazil; larissaconstant@biof.ufrj.br (L.E.C.C.); frenselg@gmail.com (G.B.F.); celimar@biof.ufrj.br (C.C.S.); diegoallonso@pharma.ufrj.br (D.A.); 6Faculdade de Farmácia, Universidade Federal do Rio de Janeiro, Ilha do Fundão, CCS, Rio de Janeiro 21941-902, RJ, Brazil; alaurambrand@gmail.com; 7Laboratório de Metabolômica Aplicada à Medicina de Sistemas (Meta2MS), Instituto de Pesquisa de Produtos Naturais Walter Mors, Universidade Federal do Rio de Janeiro, Ilha do Fundão, CCS, Rio de Janeiro 21941-599, RJ, Brazil; juliasantos.n@gmail.com; 8Instituto de Química, Universidade Federal do Rio de Janeiro, Rio de Janeiro 21941-599, RJ, Brazil; tlmontagnoli@iq.ufrj.br; 9Laboratório de Farmacologia Cardiovascular (LabCardio), Universidade Federal do Rio de Janeiro, Ilha do Fundão, CCS, Bloco J—Sala J1-11, Rio de Janeiro 21941-902, RJ, Brazil; gsudo@icb.ufrj.br; 10Programa de Pós-Graduação em Farmacologia e Química Medicinal, Universidade Federal do Rio de Janeiro, Ilha do Fundão, CCS, Rio de Janeiro 21941-902, RJ, Brazil

**Keywords:** in vitro assays, SARS-CoV-2 inhibition, main protease (M^pro^), papain-like protease (PL^pro^), protease inhibition, molecular docking

## Abstract

Endemic and pandemic viruses represent significant public health challenges, leading to substantial morbidity and mortality over time. The COVID-19 pandemic has underscored the urgent need for the development and discovery of new, potent antiviral agents. In this study, we present the synthesis and anti-SARS-CoV-2 activity of a series of benzocarbazoledinones, assessed using cell-based screening assays. Our results indicate that four compounds (**4a**, **4b**, **4d**, and **4i**) exhibit EC50 values below 4 μM without cytotoxic effects in Calu-3 cells. Mechanistic investigations focused on the inhibition of the SARS-CoV-2 main protease (Mpro) and papain-like protease (PLpro) have used enzymatic assays. Notably, compounds **4a** and **4b** showed Mpro inhibition activity with IC50 values of 0.11 ± 0.05 and 0.37 ± 0.05 µM, respectively. Furthermore, in silico molecular docking, physicochemical, and pharmacokinetic studies were conducted to validate the mechanism and assess bioavailability. Compound **4a** was selected for preliminary drug-likeness analysis and in vivo pharmacokinetics investigations, which yielded promising results and corroborated the in vitro and in silico findings, reinforcing its potential as an anti-SARS-CoV-2 lead compound.

## 1. Introduction

In the last 2 decades, global humanity has faced several viral outbreaks that represented public health challenges and affected human health and safety [1]. Coronavirus disease 2019 (COVID-19) affected millions of people, with a devastating impact on lives, livelihoods, and the global economy [2,3]. The last six major viral pandemics comprised respiratory viruses, three of them were caused by coronaviruses: severe acute respiratory syndrome (SARS-CoV-1) in 2002–2004, Middle Eastern respiratory syndrome (MERS) in 2012, and SARS-CoV-2 (COVID-19) in 2019, followed by influenza A (H1N1) viruses in 2009 (swine flu), Ebola in 2013–2016, and Zika virus infections in 2015 [4].

Coronaviruses (CoVs) are enveloped, single-stranded, positive-sense RNA viruses belonging to the order *Nidovirales*, family *Coronaviridae*, and subfamily *Coronavirinae*. They can be classified as alpha, beta, gamma, and delta coronaviruses [5]. SARS-CoV-2 is classified as a beta coronavirus. The life cycle of the virus begins after its entry into the human cell, because of the interaction between the virus spike protein (spike glycoprotein) and the ACE2 receptor (angiotensin-converting enzyme 2) on host cells. Then, the virus starts its replication process, which includes genome translation and replication that depend on the virus-encoded cysteine proteases main protease (M^pro^) and papain-like protease (PL^pro^) to be successful [6]. SARS-CoV-2 can infect several organs and tissues, causing systemic disorders ranging from flu-like symptoms to the death of the patient [7].

Even though the infection rate has declined due to vaccination, the effective vaccines against coronavirus do not provide full immunity to humans against the virus, they just alleviate the severity of symptoms and reduce the risk of death, not preventing viral action in many cases [8]. This fact, together with the high ability of the virus to mutate, emphasizes the need for effective antiviral drugs [9]. In this context, research to discover novel available small molecules that are also selective for SARS-CoV-2 targets is necessary for advances in this disease.

Natural products (NPs) and their analogs have historically been a source of privileged structures for bioactive compounds to treat different human diseases [10,11,12]. Carbazoles constitute an important class of indole alkaloids containing three cycles with a five-membered ring with a nitrogen atom in the center, fused to two six-membered aromatic rings on each side [13]. This class of natural products has been investigated for its biological properties, including antiviral activity [14]. O-Methylmukonal (**1**, Figure 1) is an example of isolated carbazole with antiviral activity for HIV-1 with EC_50_ = 12 μM [15]. Carprofen (**2**, Figure 1) is a carbazole that was used as an anti-inflammatory [16] and has been highlighted as a potential inhibitor of the SARS-CoV-2 M^pro^ [17]. 

Another intriguing group of compounds is the naphthoquinones, which are quinones related to the naphthalene system [18]. These compounds feature a benzene ring fused with a quinone structure, which consists of cyclic diones conjugated with two carbonyl groups located at the 1,2 or 1,4 positions [19]. Santos et al. (2020) [20] performed a virtual screen of 688 naphthoquinones for SARS-CoV-2 Mpro, from which 24 were selected and evaluated for M^pro^ and Pl^pro^ enzymatic activity, the most active (**3**) are shown in Figure 1. 

As an extension of our efforts to develop novel synthetic natural product derivatives with biological activity [21,22,23], we present herein the synthesis and antiviral activity of a series of benzocarbazoledinones (**4**, Figure 1) that have antibiotic and/or antitumoral activity described [24,25,26]. Therefore, we investigated the possible mechanism of inhibition based on the inhibition of SARS-CoV-2 M^pro^ and PL^pro^ and used in silico molecular docking, physicochemical, and pharmacokinetic studies for the validation of the mechanism and other bioavailability.

## 2. Materials and Methods

The reagents and solvents were obtained commercially (Aldrich) and used without further purification. All reactions that required heating were performed using an oil bath. NMR spectra were recorded in deuterated chloroform, acetone, or dimethyl sulfoxide, with tetramethylsilane (TMS) as the internal standard. The samples were analyzed at 400 and 500 MHz ^1^H NMR and 101 and 126 MHz ^13^C NMR using a Varian Unity. Chemical shifts are reported as δ values (ppm). The following abbreviations are used for the multiplicities: s: singlet, d: doublet, dd: doublet of doublet, t: triplet, q: quadruplet, m: multiplet. Melting points were uncorrected. The progress of reactions was monitored by thin-layer chromatography (TLC) or ^1^H NMR and column chromatography was carried out on silica gel; flash column chromatography silica gel 60 (40–60 mm) was employed. Analytical TLC was conducted using ALUGRAM^®^ Xtra SIL G/UV_254_ silica gel plates, and the spots were determined under UV light (λ = 254 nm and 365 nm). High-resolution mass spectra (HRMS) were obtained with a Solarix XR mass spectrometer with an Electrospray Ionization (ESI) source coupled to a Fourier Transform-Ion Cyclotron Resonance (FT-ICR) mass analyzer.

### 2.1. Chemistry

General procedure for the synthesis of benzocarbazoledione derivatives (**4a**–**i**). In a reaction tube was added a suspension of Pd(OAc)_2_ (0.0028 g, 0.01 mmol), Ag_2_O (0.0345 g, 0.15 mmol), K_2_CO_3_ (0.0345 g, 0.25 mmol), and 2-N-phenyl-aminonaphtoquinone (**9a**–**i**) (0.1 mmol) [27,28,29,30] in mixture AcOH-PivOH (3:1), which was heated for 24 h at 120 °C. Then, the mixture was allowed to cool to rt, diluted in AcOEt, filtered in celite, and concentrated under reduced pressure. The crude material was purified using a silica gel column with dichloromethane as the solvent.

5H-Benzo[b]carbazole-6,11-dione (**4a**): red solid (22 mg, 88% yield). ^1^H NMR (500 MHz, DMSO-d_6_) δ (ppm): δ 13.02 (s, 1H), 8.17 (d, J = 8.0 Hz, 1H), 8.06 (td, J = 7.8, 1.4 Hz, 2H), 7.79 (dtd, J = 22.6, 7.4, 1.5 Hz, 2H), 7.56 (d, J = 8.2 Hz, 1H), 7.45–7.37 (m, 1H), 7.33 (t, J = 7.5 Hz, 1H). ^13^C NMR (126 MHz, DMSO-d_6_) δ (ppm): δ 180.75, 177.97, 138.66, 137.59, 134.62, 134.51, 133.56, 133.05, 127.37, 126.49, 126.41, 124.37, 124.35, 122.81, 117.83, 114.28. 

2-Methoxy-5H-benzo[b]carbazole-6,11-dione (**4b**): red solid (24 mg, 88% yield). ^1^H NMR (500 MHz, CDCl_3_) δ (ppm): δ 9.41 (s, 1H), 8.25 (dd, J = 7.6, 1.4 Hz, 1H), 8.17 (dd, J = 7.5, 1.4 Hz, 1H), 7.81 (d, J = 2.5 Hz, 1H), 7.76 (td, J = 7.5, 1.4 Hz, 1H), 7.70 (td, J = 7.5, 1.4 Hz, 1H), 7.42 (d, J = 9.0 Hz, 1H), 7.11 (dd, J = 9.0, 2.5 Hz, 1H), 3.95 (s, 3H). ^13^C NMR (126 MHz, DMSO-d_6_) δ (ppm): δ 180.70, 177.66, 157.41, 137.46, 134.64, 134.62, 133.90, 133.59, 133.21, 126.49, 126.46, 125.31, 118.87, 117.48, 115.43, 102.59, 55.84.

2-Methyl-5H-benzo[b]carbazole-6,11-dione (**4c**): red solid (23 mg, 90% yield). ^1^H NMR (400 MHz, CDCl_3_) δ (ppm): δ 9.47 (s, 1H), 8.25 (d, J = 7.6 Hz, 1H), 8.21 (s, 1H), 8.16 (d, J = 7.3 Hz, 1H), 7.75 (t, J = 7.4 Hz, 1H), 7.69 (t, J = 7.4 Hz, 1H), 7.42 (d, J = 8.3 Hz, 1H), 7.30 (s, 1H), 2.52 (s, 3H). ^13^C NMR (126 MHz, CDCl_3_) δ (ppm): δ 181.16, 178.15, 136.68, 135.95, 134.81, 134.39, 134.11, 132.82, 129.64, 126.83, 126.27, 124.95, 123.03, 118.43, 112.39, 110.00, 21.60.

4-Methoxy-5H-benzo[b]carbazole-6,11-dione (**4d**): red solid (25 mg, 92% yield). ^1^H NMR (500 MHz, CDCl_3_) δ (ppm): δ 9.64 (s, 1H), 8.25 (d, J = 7.6 Hz, 1H), 8.17 (d, J = 7.5 Hz, 1H), 7.94 (d, J = 8.1 Hz, 1H), 7.72 (dt, J = 27.9, 7.5 Hz, 2H), 7.29 (d, J = 8.0 Hz, 1H), 6.84 (d, J = 7.8 Hz, 1H), 4.00 (s, 3H). ^13^C NMR (101 MHz, CDCl_3_) δ (ppm): δ 181.14, 178.01, 171.18, 146.88, 136.17, 134.73, 134.09, 132.82, 128.80, 126.81, 126.32, 125.79, 125.11, 119.13, 115.60, 106.53, 55.62.

2,4-Dimethoxy-5H-benzo[b]carbazole-6,11-dione (**4e**): red solid (27 mg, 90% yield). ^1^H NMR (500 MHz, CDCl_3_) δ (ppm): δ 9.41 (s, 1H), 8.27 (dd, J = 7.7, 1.4 Hz, 1H), 8.12 (dd, J = 7.6, 1.5 Hz, 1H), 7.72 (td, J = 7.5, 1.5 Hz, 1H), 7.66 (td, J = 7.5, 1.4 Hz, 1H), 6.51 (d, J = 2.0 Hz, 1H), 6.39 (d, J = 2.0 Hz, 1H), 4.05 (s, 3H), 3.89 (s, 3H). ^13^C NMR (126 MHz, CDCl_3_) δ (ppm): δ 181.81, 177.74, 161.84, 135.06, 133.97, 132.54, 132.18, 127.44, 126.63, 125.58, 119.55, 110.33, 106.24, 105.51, 96.01, 86.59, 56.02, 55.69. HRMS (ESI): *m*/*z* calcd. For C_18_H_12_NO_4_^−^ 306.0772; found: 306.0765.

Ethyl 6,11-dioxo-6,11-dihydro-5H-benzo[b]carbazole-2-carboxylate (**4f**): red solid (24 mg, 75% yield). ^1^H NMR (500 MHz, CDCl_3_) δ (ppm): δ 9.77 (s, 1H), 9.16–9.11 (m, 1H), 8.28 (dd, J = 7.7, 1.4 Hz, 1H), 8.18 (td, J = 8.5, 1.5 Hz, 2H), 7.81–7.72 (m, 2H), 7.57 (d, J = 8.8 Hz, 1H), 4.45 (q, J = 7.1 Hz, 2H), 1.46 (s, 3H). ^13^C NMR (126 MHz, CDCl_3_) δ (ppm): δ 184.07, 181.76, 165.78, 143.43, 141.80, 135.10, 132.93, 132.68, 131.40, 131.30, 130.21, 126.71, 126.27, 120.80, 116.83, 105.19, 61.12, 14.34. HRMS (ESI): *m*/*z* calcd. for C_19_H_12_NO_4_^−^ 318.0772; found: 318.0802. 

2-Fluoro-5H-benzo[b]carbazole-6,11-dione (**4g**): red solid (18 mg, 70% yield). ^1^H NMR (400 MHz, (CD_3_)_2_CO) δ (ppm): δ 13.18 (s, 1H), 8.07 (dd, J = 7.3, 2.8 Hz, 2H), 7.92–7.77 (m, 3H), 7.60 (ddd, J = 9.2, 4.6, 2.0 Hz, 1H), 7.31 (td, J = 9.2, 2.6 Hz, 1H). ^13^C NMR (126 MHz, CDCl_3_) δ (ppm): δ 180.54, 177.80, 161.11, 158.73, 138.80, 135.29, 134.80, 133.75, 133.61 (d, J = 137.6 Hz), 126.54, 124.67, 124.56, 117.69 (d, J = 5.2 Hz), 107.12 (d, J = 3.6 Hz), 106.88 (d, J = 3.6 Hz).

7H-Dibenzo[a,h]carbazole-7,12(13H)-dione (**4i**): brown solid (28 mg, 95% yield). ^1^H NMR (500 MHz, DMSO-d_6_) δ (ppm): δ 8.80 (d, J = 8.1 Hz, 1H), 8.23 (d, J = 8.7 Hz, 1H), 8.15 (q, J = 2.7 Hz, 2H), 8.04 (d, J = 8.0 Hz, 1H), 7.90–7.78 (m, 3H), 7.68 (t, J = 7.6 Hz, 1H), 7.62 (t, J = 7.4 Hz, 1H). ^13^C NMR (126 MHz, DMSO-d_6_) δ (ppm): δ 181.46, 177.12, 136.01, 135.21, 134.49, 134.43, 133.87, 133.45, 132.59, 129.27, 127.40, 127.19, 126.58, 125.89, 122.87, 122.53, 121.33, 120.67, 119.29, 110.00. HRMS (ESI): *m*/*z* calcd. for C_20_H_10_NO_2_^−^ 296.0717; found: 296.0709.

### 2.2. Biological Activity

#### 2.2.1. Calu-3 Cytotoxicity Assay

Calu-3 cells, a submucosal gland cell line derived from a bronchial adenocarcinoma (generously provided by the Farmanguinhos platform RPT11M), were cultured in 96-well plates at a density of 1.5 × 10^4^ cells per well. The cells were treated with compounds **4a**, **4b**, **4c**, **4d**, **4e**, **4f**, **4g**, and **4i** at concentrations of 10 and 100 µM for 72 h to assess cell viability. Following the treatment period, the cells were washed with PBS (phosphate-buffered saline) and stained with a methylene blue solution containing HBSS (Hank’s Balanced Salt Solution), 1.25% glutaraldehyde, and 0.6% methylene blue for 1 h. The cells were then washed again, and an elution solution (50% ethanol, 49% PBS, and 1% acetic acid) was added for 15 min at room temperature. The supernatant was collected, and absorbance was measured using a spectrophotometer at 660 nm.

#### 2.2.2. Antiviral Activity

Calu-3 cells (1.5 × 10^4^ cells per well) were infected with the SARS-CoV-2 B.1 lineage isolate (GenBank MT710714, SisGen AC58AE2) at a multiplicity of infection (MOI) of 0.01 for 1 h at 37 °C in a 5% CO_2_ atmosphere. Following infection, the analyzed compounds were added at 10 µM or various concentrations (0.6, 1.3, 2.5, 5, and 10 µM) for 24 h. The supernatant was then collected, and viral growth was quantified using a plaque-forming unit (PFU) assay. In brief, Vero E6 cells (African green monkey kidney cells, ATCC CRL-1586) were seeded at 1.5 × 10^4^ cells per well and incubated with 50 µL of supernatant at serial dilutions (1:100 to 1:12,800) for 1 h at 37 °C in 5% CO_2_. Subsequently, 50 µL of carboxymethylcellulose medium (comprising DMEM-HG 10×, 2.4% carboxymethylcellulose, and 2% fetal bovine serum) was added, and the cells were cultured for 72 h. Cells were then fixed with 4% formalin for 3 h, stained with 0.04% crystal violet for 1 h, and PFUs were counted to determine viral titers. The CC_50_ and EC_50_ values of atazanavir served as experimental controls for assessing viral inhibition and cell viability in Calu-3 assays [31]. All viral manipulations were conducted in a biosafety level 3 (BSL3) facility, adhering to WHO guidelines [32].

The compounds used in these in vitro experiments were dissolved in 100% dimethyl sulfoxide (DMSO), aliquoted, and stored at −20 °C to prevent degradation. The final DMSO concentrations were maintained at or below 1% (*v*/*v*) in Dulbecco’s Modified Eagle Medium (DMEM), which did not affect cell growth [33,34].

#### 2.2.3. Protease Inhibition

Recombinant SARS-CoV-2 M^pro^ and PL^pro^, expressed in *E. coli* BL21(DE3)pLysS and BL21(DE3) cells, respectively, were used in a fluorescent resonance energy transfer (FRET) assay using the peptide DABCYL-AVLQ↓SGFRLL-EDANS as a substrate for M^pro^ and the peptide DABCYL-ALKG↓GKIV-EDANS for PL^pro^. The M^pro^ concentration was fixed at 1.5 μM, the substrate at 50 μM, and the compounds ranged in concentration from 0.01 to 1000 μM. The mixture was incubated in 5 mM NaCl, 20 mM Tris.HCl pH 8.0, and 5 mM DTT for 15 min at 37 °C prior to starting with the substrate. For PL^pro^, a similar protocol was applied but the enzyme was set in 1 μM and the reaction buffer was 150 mM NaCl, 20 mM Tris.HCl pH 8.0, and 5 mM DTT. EDANS emission fluorescence was monitored at each 30 s for 45 min, at 37 °C (λexc = 330 nm, λem = 490 nm). Fluorescence data (RFU) were converted into substrate cleavage specific activity using a fluorescent conversion factor (FEC) previously calculated based on the EDANS-DABCYL fluorophore pair. Maximum enzyme activity was considered in the situation with vehicle (DMSO) and the values were used to calculate the enzyme inhibition by the compounds. The concentration that inhibited 50% of the enzyme activity (IC_50_) was calculated in the software GraphPad Prism 9.0.

#### 2.2.4. Statistical Analysis

Graphs were generated using GraphPad Prism 10.0 software. One-way ANOVA followed by Dunnett’s post hoc test was employed to analyze the differences between treatment groups. Statistical significance was denoted as *** *p* ≤ 0.001. The half-maximal effective concentration (EC_50_) was determined by nonlinear regression of the log(inhibitor) versus the normalized response curve, with R^2^ values ≥ 0.9 considered acceptable. All experiments were conducted in duplicate with three technical replicates each (*n* = 6).

#### 2.2.5. Aqueous Solubility of **4a**

The aqueous solubility of compound **4a** was determined using a NanoDrop One/One^c^ Microvolume UV-Vis Spectrophotometer (ThermoFisher, Waltham, MA, USA) following the methodology described by Schneider et al. [35]. Methanol (HPLC grade) for the calibration analytical curve was purchased from Biograde, and high purity water (18.2 MΩ·cm) from a Millipore Milli-Q purification system was used. The aqueous solubility was calculated from a saturated solution of compound **4a** and quantified based on a standard curve prepared in methanol ranging from 0.02 to 400 µg·mL^−1^. For quantification, solutions of **4a** were prepared from 0.0005 to 10 µg·mL^−1^. 

#### 2.2.6. In Vivo Pharmacokinetics of **4a**

Experiments were conducted by the Animal Care and Use Committee at Universidade Federal do Rio de Janeiro (license 045/22, date of approval 8 September 2023). Male Wistar rats (280–320 g, *n* = 4) were housed at 24 °C under a 12 h light/12 h dark cycle with free access to food and water. Compound **4a** was dissolved to 15 mg/mL in DMSO (Sigma-Aldrich; St Louis, MO, USA) and intraperitoneally administered at 5 mg/kg per animal. Blood samples (500 μL) were collected at 5, 15, 30, 45, 60, 90, 120 min, and 24 h via puncture of the lateral tail vein into microtubes containing 20 mg (50 μmol) disodium EDTA. Samples were gently homogenized and centrifuged at 2000× *g* for 1 min, and each decanted plasma sample obtained was separated and frozen at −80 °C until analysis.

Extraction of **4a** was performed into a 1.5 mL plastic tube using 30 µL plasma and 120 µL of ice-cold MeOH with an internal standard (caffeine 5 µg·mL) and homogenized for 10 s. The samples were put into an ultrasonic bath for 10 min. Then, the samples were incubated at −20 °C in a freezer for 15 min for protein precipitation, followed by centrifugation at 10,000 RPM for 15 min at 10 °C. The supernatant was collected (120 µL) and transferred to a new plastic tube that was used for total evaporation in nitrogen. The concentrated extracts were reconstituted in 60 µL of water/methanol (7:3), homogenized, and transferred to a 2 mL vial with an insert for LC-MS/MS analysis.

The development of the analytical and instrument conditions for the quantitative analysis of compound **4a** in plasma was performed via liquid chromatography–tandem mass spectrometry (LC-MS/MS) using a Thermo Scientific Dionex UltiMate 3000 UHPLC liquid chromatography system (Thermo Fisher Scientific, Waltham, MA, USA) coupled to a TSQ Quantiva Triple Quadrupole Stage Mass Spectrometer, using electrospray ionization (ESI) and equipped with a degasser and TriPlus RSH autosampler (Thermo Fisher Scientific, Waltham, MA, USA). The LC separation was conducted on a C18 Hypersil Gold (100 × 2.1 mm × 3.0 µm particle diameter) (Thermo Fisher) maintained at 40 °C. Elution was performed using Milli-Q water containing 0.1% formic acid (A) and methanol containing 0.1% formic acid (B), and 5 µL of samples and controls were injected. Gradient elution at a flow rate of 0.350 mL.min^−1^ was performed as follows: 0–1 min 20% B; 1–2 min 40% B; 4.5–7.5 min 98% B; 7.6–9 min 20%. The total analysis time per sample was 9 min.

The electrospray source was operated in positive ionization mode (ESI+) at 3500 V, sheath gas flow 40 a.u., auxiliary gas flow 10 a.u., sweep gas flow 1 a.u., vaporizer heater temperature 400 °C, and ion transfer tube temp 350 °C. Selected reaction monitoring modes (SRMs) of *m*/*z* 248.7 > 165.0 (RT 4.78) and *m*/*z* 195.0 > 138.07 (TR 2.48) were used for quantification of compound **4a** and caffeine (used as an Internal Standard, IS), respectively. The data were acquired, then data processed using TraceFinder 4.1 (Thermo Fisher Scientific). Plasma pharmacokinetic parameters of **4a** were calculated by noncompartmental analysis using Prism 9 and expressed as mean ± standard error [36]. 

### 2.3. Computational Modeling

#### 2.3.1. Molecular Dynamic Simulations

To acquire the structure of SARS-CoV-2’s main protease (M^pro^) for molecular docking analyses, we utilized comparative modeling with the PDB6Y2E and PDB6Y2G models accessible in the Protein Data Bank. After obtaining the base model, we completed the C-terminal part of the protein chains to reach the main protease dimer using Modeller v9.23 software.

The protein was then prepared via the CHARMM-GUI online server. The catalytic residues CYS145 and HIS41 were protonated based on their respective pKa values, calculated using PROPKA v3, yielding CYS-H and HSD forms, respectively. The protonated protein was placed in a simulation box with the TIP3P water model, with a 15 Å distance from the solute to the box edge. A concentration of 150 mM NaCl was added to neutralize the system. An energy minimization simulation was conducted utilizing the steep descent method, with 50,000 steps to eliminate local atomic collisions in the system. Following the minimization, optimization, heating, equilibrium, and production stages were performed. The latter step employed an isothermal–isobaric ensemble from the leapfrog algorithm with an integration step of 2fs. The obtained conformations were grouped based on the set of trajectories to identify the preferred states and the conformation with the lowest overall energy content.

#### 2.3.2. Protein and Ligand Preparation

##### Compound Preparation

The studied compounds were designed using Chemsketch v12.01 software, and their three-dimensional structures were optimized using Avogadro v1.2.0 software with the MMFF94S force field. The energy of minimization was performed using the Steepest Descent algorithm, with 10,000 steps and a convergence threshold of 10 × 10^−7^. Subsequently, the protonation states of the compounds were predicted using the epik v4.6012 program from Schrödinger software (version 4.6012) at pH 7.4.

For the AutoDock Vina assays, AutoDock Tools (ADT) v1.5.6 software was employed to assign Gasteiger charges and define molecular torsions. As for the DockThor web program assays, the ligands obtained with epik were combined into a single set and subjected to docking tests using the MMFF94S force field.

##### Physicochemical Properties Analysis

We analyzed the Lipinski and Veber parameters for each substance to evaluate the most important physicochemical properties of the compounds. Additionally, we utilized the Molinspiration property engine (v2018.10) and Molinspiration bioactivity score (v2018.03), available at https://www.molinspiration.com/cgi-bin/properties (accessed on 2 May 2024), and the Swiss-ADME software (v2018.10), available at http://www.swissadme.ch/index.php (accessed on 2 May 2024). In addition, the probability of binding to plasma proteins (BPP) and bioavailability were assessed using Deep-PK [37], available at https://biosig.lab.uq.edu.au/deeppk/ (accessed on 2 May 2024). 

##### Protein Preparation

After obtaining the global minimum and centroid structures from molecular dynamics simulations of the SARS-CoV-2 M^pro^, docking assays were performed between the different compounds and the target conformational state protein. Orthorhombic grids were defined to simulate protein–ligand interactions in chains A and B, considering different centers of mass (Table 1) and 30 Å of distance.

After the identification of the catalytic residues H41 and C145, the center of the box on each chain was determined by reference to the center of the sulfur atom bond of the cysteine residue (C145: SG-) closest to the hydrogen bond to the ring nitrogen of the histidine residue (H41:ND1 or H41:NE2). Table 1 displays the centers of the grids.

#### 2.3.3. Molecular Docking

The AutoDock Vina and DockThor programs were employed to perform docking studies, enabling the investigation of potential interactions between the compounds and the target protein. The protein structures obtained from molecular dynamics simulations served as the basis for these docking assays, providing insights into the binding modes and potential binding sites of the compounds.

##### Analysis of the Best Compounds

The Z-score function was utilized to assess the molecular docking results using Equation (1).
Z = (Xi − X)/σ(1)

The function’s components include (Xi), which represents the average affinity energy for each evaluated compound, X, which denotes the overall average energy between compounds, and σ, which signifies the standard deviation of energy values for each molecule. 

## 3. Results and Discussion

These benzocarbazoledinone derivatives were designed through the fusion of the carbazole and benzoquinone groups, two pharmacophoric chemical entities together into a unified framework. The analysis of the substituent in the aromatic will provide the physicochemical properties that can determine the distinct contributions of each fragment to the desired activity profile of these compounds (**4a**–**i**, Figure 2). 

### 3.1. Synthesis

The synthesis of benzocarbazoledinone derivatives (**4a**–**i**) followed the general pathway outlined in Figure 1. 

The first step was the reaction of 2-bromo-1,4-naphthoquinone (**5**) with benzylamine to generate 2-benzylamino-1,4-naphthoquinone (**6**) with 92% yield, followed by the hydrogenolysis reaction to generate 2-amino-1,4-naphthoquinone (**7**) with a quantitative yield [27,28]. So, we investigated the palladium-catalyzed coupling reaction between the 2-amino-1,4-naphthoquinone (**7**) and phenyl halides (**8a**–**8i**) to form 2-N-phenyl-amino-1,4-naphthoquinone (**9a**–**9i**) (Figure 1), the intermediaries of the final products of interest. The reaction was conducted using Pd(OAc)_2_ (5 mol%), as the palladium font, and XPhos (10 mol%), as the ligand, in toluene under microwave conditions. The products presented moderated to good yields (up to 79%). After that, we investigated the intramolecular C-H activation reaction using the 2-N-phenyl-amino-1,4-naphthoquinone (**9a**–**9i**) in the presence of Pd(OAc)_2_ and K_2_CO_3_ in a mixture of AcOH:PivOH (3:1) as a solvent to lead the corresponding benzocarbazolequinones (**4a**–**4i**) with good yields (70–96%). All the experiments were carried out with the addition of Ag_2_O as an oxidant agent. This catalytic process does not directly depend on the influence of the substituents EWGs or EDGs. A negative effect for the catalytic process was clearly observed in the presence of the pyridine ring at **9j**; the reaction did not work. These eight compounds were directed for the anti-SARS-CoV-2 in vitro activity.

The structures of all the prepared compounds were confirmed by spectral analyses and ^1^H, ^13^C NMR, and the obtained data were in full agreement with the proposed structures.

### 3.2. Biological Activity Evaluation

The studies were conducted using the synthetic compounds **4a**–**i**. To evaluate the compounds’ effects in an in vitro cell model, we first investigated their cytotoxicity in a human type II pneumocyte cell model (Calu-3 cells) using the methylene blue assay. Our results demonstrated that most benzocarbazoledinones exhibited no toxicity in Calu-3 cells at any of the concentrations tested (Figure 3), except for compound **4i**, which showed moderate toxicity at 100 µM, with 68% cell viability (Figure 3b). However, this concentration was ten times higher than those used in the antiviral assay, suggesting that compound **4i** is safe for the cell model used, allowing us to proceed with further investigation.

The antiviral activity was assessed by treating Calu-3 cells with 10 µM of each compound for 24 h. We observed that only compound **4c** was ineffective in inhibiting SARS-CoV-2 replication, while the other compounds exhibited inhibition rates ranging from 47% to 96% (Figure 4). Specifically, the presence of fluorine (**4g**) or ethyl acetate (**4f**) as a substituent appeared to decrease the efficiency of inhibition (69% and 64%, respectively) when compared with compound **4a**, a benzocarbazoledinone without substituent, which showed 95% inhibition (Figure 4). Interestingly, the presence of methoxy substituents influenced the compounds’ activities differently depending on their positions on the aromatic ring. The presence of two methoxy groups (compound **4e**) reduced SARS-CoV-2 inhibition by 47.44% compared to compound **4a** (Figure 4). On the other hand, compounds with a single methoxy group showed minimal changes in activity when the substitution was at the ortho position (**4d** = 8.1% difference) or no significant change when the substitution was at the para position (**4b**) (Figure 4) compared to compound **4a**.

The most active compounds, which inhibited viral replication by more than 80% (**4a**, **4b**, **4d**, and **4i**), had their anti-SARS-CoV-2 activity (EC_50_) further analyzed (Figure 5 and Table 2). These compounds showed shallow EC_50_ values ranging from 0.7 to 3.7 μM. These EC_50_ values are comparable to those observed for FDA-approved or indicated drugs for COVID-19 treatment and other notable anti-SARS-CoV-2 molecules in the literature, such as Nirmatrelvir (EC_50_ = 0.1 µM) [38,39], Molnupiravir (EC_50_ = 1.96 µM) [40], Remdesivir (EC_50_ = 2.49 µM) [41], and Lopinavir combined with Ritonavir (EC_50_ = 5.3 µM) [42,43]. These findings suggest that the benzocarbazoledinones **4a**, **4b**, **4d**, and **4i** are promising candidates with significant anti-SARS-CoV-2 activity.

We further investigated the possible mechanism of action of the four selected compounds. Since viral replication is closely dependent on the proteolytic activity of the non-structural proteases M^pro^ and PL^pro^, inhibiting these enzymes constitutes an important strategy for antiviral therapy. The protease inhibition studies were performed using a FRET-based assay using recombinant purified SARS-CoV-2 M^pro^ and PL^pro^ enzymes. As shown in Table 3, compounds **4a** and **4b** strongly inhibited M^pro^ activity at nanomolar concentrations. These compounds were also able to inhibit PL^pro^; however, this inhibition was only achieved at higher compound concentrations. Notably, compounds **4d** and **4i** did not inhibit M^pro^ as effectively as their analogs, suggesting that these molecules may operate through alternative mechanisms of M^pro^ inhibition not explored in this study or potentially act on targets beyond the viral proteases. GC-376, a dipeptide that covalently binds to coronavirus M^pro^, was used as positive control for M^pro^ [44]. GRL-0617, a non-covalent competitive inhibitor of PL^pro^, was used as its positive control [45].

In fact, it is quite common for the concentration of a molecule required to inhibit 50% of its maximum inhibitory virus replication activity to be higher than that needed to inhibit 50% of its maximum enzymatic activity, especially if this activity is its primary mechanism of action. A very clear example of this is oseltamivir against seasonal influenza virus. This molecule can inhibit viral replication at micromolar concentrations, while it inhibits the activity of the neuraminidase enzyme, its main mechanism of action described to date, at nanomolar levels, highlighting the difference between the concentration needed for enzyme inhibition versus viral replication inhibition [46].

### 3.3. Computational Modeling Evaluation

#### 3.3.1. Molecular Docking Evaluation

Once the two promising compounds **4a** and **4b** had shown excellent inhibition against M^pro^, we conducted an in silico investigation to demonstrate the possible interactions in the active site of these two compounds. The SARS-CoV-2 M^pro^ is a homodimer with A and B chains, where each monomer is composed of 306 amino acids and consists of three different domains. The active site localized in the cleft between domain I and domain II is responsible for the cleavage of the polyproteins in nonstructural proteins (nsps) [47,48]. This cysteine protease has a catalytic dyad composed of the amino acids His41 and Cys145. Currently, M^pro^ is well-established as a SARS-CoV-2 target and has been the focus of many publications in the area [49,50].

The conformational analysis and pharmacophoric profile assessment through molecular docking between M^pro^ and compounds **4a** and **4b** were conducted, as shown in Figure 6. Notably, the residues present in the catalytic site engage in significant interactions. Compound **4a** (Figure 6A,B) seems to make polar interactions with the residue H164. Additionally, other interactions were identified in the two-dimensional analysis, including polar interactions with the residues Q189, T190, and T25; negative charged interactions with E166 and D187; hydrophobic interactions with L27; and finally, hydrogen bonds with C145, H41, and N142. Compound **4b** exhibited the same interactions as the previous one.

When comparing the best results for the compounds of interest with the control compounds Atazanavir and Nirmatrelvir, similar interactions with the target molecules are observed, such that the catalytic residues of the active site also form analogous bonds. For instance, Nirmatrelvir (Figure 7) formed hydrogen bonds with C145, H163, F140, L141, S144, and D187. Similarly, the control Atazanavir also engaged in hydrogen bond interactions with the residues D187, Q198, and H41 (histidine is in its protonated form HSD). 

The study utilized molecular docking software, AutoDockVina (version 1.1.2) and DockThor (version 2.0). The methodology involved assessing the affinity energies and interactions between ligand compounds and a target protein (SARS-CoV-2 M^pro^). The compounds **4a** and **4b** presented the following average affinity to M^pro^: −7.644 kcal/mol and −7.660 kcal/mol, respectively. Atazanavir and Nirmatrelvir were used as controls and obtained average affinities of −5.700 kcal/mol and −6.600 kcal/mol, respectively. 

Nirmatrelvir and Atazanavir were used in this study as positive controls, given that both have scientific backing as protease inhibitors. Nirmatrelvir was chosen as one of the in silico controls because it is a major component of Paxlovid, an oral antiviral drug approved for COVID-19 treatment, with clinically proven efficacy as a M^pro^ inhibitor. It demonstrates potent activity against SARS-CoV-2 replication by interacting with the C145 residue, promoting the cleavage of the H41-C145 catalytic dyad [51].

Atazanavir, although not clinically used for COVID-19 treatment, is supported by the literature for its activity against M^pro^. It operates through a slightly different mechanism, as it is a competitive inhibitor of this protease and requires a water molecule to enhance interactions with the H41 residue at the catalytic site [31]. Furthermore, other studies have shown that Atazanavir can form hydrogen bonds with the catalytic amino acids, corroborating the docking results of the present study [43]. While both controls interact with the catalytic residues, despite their experimentally validated efficacy, they did not simultaneously interact with the residues at the catalytic site in the in silico experiments. This finding positions molecules **4a** and **4b** (Figure 6) as promising and potent inhibitors of SARS-CoV-2. 

Table 4 presents the molecular docking results for all the compounds, showing the respective distances in Ångströms between the molecules and the C145 sulfur atom in the Mpro active site, as well as the corresponding cKi values (μM) and affinity energies (kcal/mol).

#### 3.3.2. In Silico Prediction of Drug-Likeness, Physicochemical Properties, and Pharmacokinetic Profile

To evaluate compounds in silico, it is crucial to predict their physical and chemical properties using specialized software. For this purpose, we applied the Lipinski and Veber criteria in conjunction with the Swiss-ADME and Molinspiration servers. For a compound to be considered viable to be a drug candidate, it must satisfy essential criteria such as oil–water partitioning, polarity, and molar mass [53,54]. Failing to meet these parameters can compromise absorption and permeability.

In this regard, to effectively predict the molecule’s suitability, it should not violate more than two parameters of Lipinski’s Rule of Five (RO5) [55,56]. These parameters include the oil/water partition coefficient (LogPw/o), which must not exceed 5.00, and the molecule’s molar mass, which should be less than 500 g/mol. Additionally, the molecule should contain no more than ten groups that allow hydrogen bonds (determined by the total sum of nitrogen and oxygen atoms) and no more than five hydrogen donors (sum of OH and NH groups). Furthermore, it must meet certain additional criteria, such as the number of rotatable bonds (nrotbs), which cannot exceed ten, and the polar surface area (PSA), which cannot exceed 140 Å^2^. An alternative approach to assessing the polar surface area is by using the Topological Polar Surface Area (TPSA), which evaluates the exposure of areas with the highest potential for forming hydrogen bonds. A higher TPSA indicates a greater likelihood of the molecule interacting with biological targets in a polar manner. 

The Molinspiration and Swiss-ADME software (v2018.10) analyses revealed that Lipinski’s rules were not violated by either molecule. Refer to Appendix A for detailed outcomes about the individual molecules and controls utilized. 

Lipinski’s rule predicts certain physicochemical molecule parameters but does not account for the intricate drug absorption, distribution, metabolism, excretion, and toxicity processes. Lipinski’s RO5 and Veber’s parameters identify viable drug candidates for in vitro testing, suggesting that these compounds may possess improved bioavailability.

It is well established that the majority of drugs have the capacity to bind to plasma proteins, including albumin, α1-acid glycoprotein, lipoproteins, and α-, β-, and γ-globulins. It should be noted that only the unbound fraction of drugs is distributed from the blood to the tissues and sites of action, as well as being metabolized and excreted. The bound and unbound fractions maintain a dynamic equilibrium, whereby as the free fraction is metabolized and excreted, the previously bound molecules dissociate from the plasma proteins, thereby becoming available to exert pharmacological and/or toxicological action [57,58].

To ascertain the probability of molecules **4a** and **4b** being in an unbound state or complexed with plasma proteins, Deep-PK predictions, a deep learning-based tool for predicting ADMET properties, was employed. The Deep-PK Predictions software indicated that compound **4a** had an unbound fraction of 42.41%, while for compound **4b** the unbound fraction was 36.65%. To assess the accuracy and reliability of the software, the reference compounds atazanavir and nirmatrelvir were also subjected to analysis. The unbound fraction predicted for atazanavir was 12.96%, which is consistent with the experimental data indicating a 14% free fraction [59]. In the case of nirmatrelvir, the predicted free fraction was 23.6%, while experimental data indicate a free fraction of 30% [60]. These findings indicate that Deep-PK Predictions is an effective tool for predicting the free fraction rates with reasonable accuracy for the compounds under evaluation, with results that align well with experimental data. The elevated free fractions of compounds **4a** and **4b** suggests substantial accessibility for distribution to tissues and pharmacological action, as well as rapid metabolization and excretion. This favors a more efficient pharmacokinetic profile and potential therapeutic response.

### 3.4. Drug-Likeness Properties and In Vivo Pharmacokinetics

Considering the aim to identify a promising anti-SARS-CoV-2 lead compound, the analysis of Table 5 allowed the selection of compound **4a** for preliminary drug-likeness analysis.

The experimental aqueous solubility of compound **4a** was determined and indicated a favorable hydrolipidic profile (Appendix A). 

Determination of the solubility and prediction of drug-likeness properties of **4a** prompted the investigation of its in vivo pharmacokinetics (Table 5 and Figure 8). When **4a** was administered at 5 mg/kg i.p., its plasma concentration reached a maximum (C_MAX_) of 0.77 ± 0.12 μM after 20 ± 5 min, indicating fast absorption due to excellent permeation of biological membranes. After 24 h, molecule **4a** was still detected in rat plasma at a concentration of 0.12 ± 0.04 µM, closely aligning with its M^pro^ IC_50_ of 0.11 ± 0.05 µM. Additionally, this plasma concentration is on the same scale as the EC_50_ value of 0.75 ± 0.1 µM against SARS-CoV-2, highlighting that the levels achieved in vivo are within a range consistent with its in vitro antiviral activity, underscoring its potential efficacy in therapeutic settings. This profile results from a reduced clearance (CL/F) of **4a** due to extensive partitioning into tissues and slower elimination kinetics, as indicated by the apparent volume of distribution (V_z_/F) of 98 L/kg and terminal half-life (t_1/2,z_) of 37 h, respectively (Table 5).

## 4. Conclusions

This study not only supports the protease Mpro as a viable target for benzocarbazoledinones against SARS-CoV-2 but also presents evidence that these compounds, without substituents and containing methoxy groups in para-positions of the nitrogen, exhibit promising potential for inhibiting SARS-CoV-2 proliferation. These compounds were successfully synthesized with moderate to high yields (70–90%) and demonstrated the effectiveness of the synthetic route. Among them, four compounds (**4a**, **4b**, **4d**, and **4i**) exhibited not only high activity for inhibiting the proliferation of the virus but also favorable selectivity indexes (SI > 50 μM). Additionally, molecular docking confirmed the conformations and interactions with the Mpro and predictions regarding drug-likeness and pharmacokinetic profiles indicated drug-like characteristics and favorable pharmacokinetic properties, which were confirmed in vivo after a single intraperitoneal dose. 

This research highlights the successful design and synthesis of benzocarbazoledinones with potent antiviral activities. The combination of cell-based screening, in vitro enzymatic testing, and in silico analysis has provided valuable insights into the structure–activity relationships and druggability of these compounds. These findings contribute to the ongoing efforts in the development of effective treatments against coronaviruses. Further investigations and optimization of these benzocarbazoledinones hold promising prospects for the development of novel antiviral therapeutics.

## Data Availability

Data is contained within the article or Appendix A.

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
