# Peer review of "Benzocarbazoledinones as SARS-CoV-2 Replication Inhibitors: Synthesis, Cell-Based Studies, Enzyme Inhibition, Molecular Modeling, and Pharmacokinetics Insights"

_viruses, 2024, doi:10.3390/v16111768_

Round 1

Reviewer 1 Report

Comments and Suggestions for Authors

The study by de Souza and colleagues describing the synthesis and characterization of a series of benzocarbazoledinones as inhibitors of SARS-CoV-2 is a well designed and executed series of studies.  The authors do a very good job of describing the synthesis of these compounds and the initial characterization of their antiviral activity and mechanism of action.  I do, however, have a significant concern around the translational findings and their relevance to the clinic. 

General Comment

Specifically, the authors conducted a single PK study using the intraperitoneal dosing route.  In the conclusion section (lines 590-592), they suggest that the “..pharmacokinetic profiles indicated drug-like characteristics and favorable pharmacokinetic properties, which were confirmed in vivo after single intraperitoneal dose.”.  I would argue that this single PK study using a parenteral route of administration does little to confirm a favorable PK profile.  While the drug was absorbed from the site of injection, unless the authors plan to utilize this route of administration, this data does little to predict how the compound would behave following oral dosing.  Furthermore, on line 569 the authors suggest that the key parameter for determining antiviral activity is the Mpro IC50 of 0.11 uM.  This is in stark contrast to the in vitro EC50 against SARS-CoV-2 shown in Table 2 of 0.75 uM.  Furthermore, the authors use the IP PK data to suggest that the results compare favorably to nirmatrelvir and molnupiravir.  This is an apples to oranges comparison that should be removed.  The authors should not suggest that these compounds have “favorable” pharmacokinetics as that statement has not been validated by the data contained within the current report. 

The authors only tested a single SARS-CoV-2 isolate (B.1 lineage).  Given the wide-spread availability of the different variants and the differences in activity, it would be helpful to have the results from multiple isolates.

For antivirals, a well-known parameter associated with drug activity is protein binding as only the free drug is available to interact with the drug target.  While the authors assessed aqueous solubility, they did not try to evaluate protein binding.  This is a significant oversight and should be corrected as it has tremendous bearing on the potential clinical utility of these compounds.

Specific Comments

Line 448:  The authors suggest that compounds 4d and 4i are poor inhibitors of Mpro and PLpro which may be the result of a different mechanism of action; however, they did not follow-up with any additional studies.  Given the close, structural relationship between all of these compounds, is it possible that the compounds that did show potent inhibition of these proteases could also have more than one mechanism?  There should be some additional comment in the discussion section about this finding.  In addition, the end of the sentence on lines 448 to 450 (shown below) doesn’t make sense.  What is meant by “…indicating the strong interaction and inhibition of the Mpro.”?

Interestingly, compounds 4d and 4i poorly inhibited both the Mpro and PLpro, suggesting a different mechanism of action, indicating the strong interaction and inhibition of the Mpro.

Table 5 and Figure 8:  The authors estimate the half-life of 4a to be ~37 hours.  This does not appear to be correct.  Based on the curve shown in Figure 8, I would estimate the half-life to be <10 hours. 

Line 569:  As discussed above, the key driver of clinical utility is not the Mpro IC50 but rather the EC50 for the virus.  Given that the Cmax observed following the single IP injection was approximately the same as the EC50 (0.77 uM versus 0.75 uM, respectively), I do not believe the statement of “..suggesting prolonged antiviral activity even after a single i.p. injection.” is accurate and should be removed.

Lines 573-575: As discussed above, the PK parameters for molnupiravir and nirmatrelvir are following oral dosing while the parameters for 4a are following IP dosing.  As they are not directly comparable, this section should either be modified with the appropriate disclaimers or deleted.

Author Response

Response to referee 1 (R1)
R1: The study by de Souza and colleagues describing the synthesis and characterization of a series of benzocarbazoledinones as inhibitors of SARS-CoV-2 is a well designed and executed series of studies. The authors do a very good job of describing the synthesis of these compounds and the initial characterization of their antiviral activity and mechanism of action. I do, however, have a significant concern around the translational findings and their relevance to the clinic.
Answer: We appreciate the comments and suggestions. All questions have been addressed. Everything modified in the new manuscript version is highlighted in red.
General Comment
R1: Specifically, the authors conducted a single PK study using the intraperitoneal dosing route. In the conclusion section (lines 590-592), they suggest that the “..pharmacokinetic profiles indicated drug-like characteristics and favorable pharmacokinetic properties, which were confirmed in vivo after single intraperitoneal dose.”. I would argue that this single PK study using a parenteral route of administration does little to confirm a favorable PK profile. While the drug was absorbed from the site of injection, unless the authors plan to utilize this route of administration, this data does little to predict how the compound would behave following oral dosing.
Answer: The cited PK study design (single i.p. dose) was chosen based on two premises. First, i.p. administration is commonly used for testing the efficacy of new drug candidates in animal models as an easier and less invasive alternative to the intravenous route. Although absorption kinetics may limit bioavailability after an oral dose, this issue could be specifically addressed using different formulations. However, elimination kinetics was a major concern, since the high lipophilicity of 4a could enhance its metabolic clearance rate and limit its clinical use. The results obtained in this single dose study demonstrated that a significant amount of 4a was still found in plasma even 24 hours after i.p. administration, which was judged favorable and will prompt further studies. Second, we should remind that critically ill COVID patients remain at higher risk and should benefit of new drug therapies for improving survival and, in this context, parenteral dosing is preferred to the oral route.
R1: Furthermore, on line 569 the authors suggest that the key parameter for determining antiviral activity is the Mpro IC50 of 0.11 uM. This is in stark contrast to the in vitro EC50 against SARS-CoV-2 shown in Table 2 of 0.75 uM.
Answer: This sentence was rewritten in the text. In fact, it is quite common for the concentration of a molecule required to inhibit 50% of viral replication to be higher than
that needed to inhibit 50% of an enzymatic activity, especially if this activity is its primary mechanism of action. A very clear example of this is oseltamivir against seasonal influenza virus. This molecule can inhibit viral replication at micromolar concentrations, while it inhibits the activity of the neuraminidase enzyme, its main mechanism of action described to date, at nanomolar levels, highlighting the difference between the concentration needed for enzyme inhibition versus viral replication inhibition (Moscona, 2005).
Moscona A. Neuraminidase inhibitors for influenza. N Engl J Med. 2005 Sep 29;353(13):1363-73. doi: 10.1056/NEJMra050740. PMID: 16192481.
R1: Furthermore, the authors use the IP PK data to suggest that the results compare favorably to nirmatrelvir and molnupiravir. This is an apples to oranges comparison that should be removed. The authors should not suggest that these compounds have “favorable” pharmacokinetics as that statement has not been validated by the data contained within the current report.
Answer: We agree with this referee, as suggested, the comparison was removed from text (line 572).
R1: The authors only tested a single SARS-CoV-2 isolate (B.1 lineage). Given the wide-spread availability of the different variants and the differences in activity, it would be helpful to have the results from multiple isolates.
Answer: Thank you for this valuable suggestion. We acknowledge the importance of evaluating antiviral efficacy across multiple SARS-CoV-2 variants, especially given the differences in transmissibility and immune escape potential that have emerged over the course of the pandemic. However, protease analysis indicates a high degree of structural and functional similarity in the main protease (Mpro) across SARS-CoV-2 variants, including the B.1 lineage and more recent variants, such as Delta and Omicron (Lee JT et al., 2022; Kandwal S & Fayne D, 2023).
In our laboratory, we do have isolates of different SARS-CoV-2 variants. However, we opted to use the B.1 isolate in this study due to its robust plaque formation in Vero E6 cell cultures, which allowed us to apply the gold-standard plaque assay. In contrast, the Omicron variant, for example, does not form reproducible plaques under the same conditions, leading many research groups to rely on indirect quantification methods, such as qRT-PCR (Essaidi-Laziosi M et al., 2024). While qRT-PCR is a powerful tool, it lacks the direct, high-resolution assessment of viral infectivity that plaque assays provide, and thus may not offer the same level of precision for evaluating in vitro antiviral activity.
For these reasons, we prioritized using the B.1 isolate, which offers the most reliable results in plaque assays with Vero E6 cells, ensuring robust and reproducible findings.
-
Lee JT, Yang Q, Gribenko A, Perrin BS Jr, Zhu Y, Cardin R, Liberator PA, Anderson AS, Hao L. Genetic Surveillance of SARS-CoV-2 Mpro Reveals High Sequence and Structural Conservation Prior to the Introduction of Protease Inhibitor Paxlovid. mBio. 2022 Aug 30;13(4):e0086922. doi: 10.1128/mbio.00869-22. Epub 2022 Jul 13. PMID: 35862764; PMCID: PMC9426535.
-
Kandwal S, Fayne D. Genetic conservation across SARS-CoV-2 non-structural proteins - Insights into possible targets for treatment of future viral outbreaks. Virology. 2023 Apr;581:97-115. doi: 10.1016/j.virol.2023.02.011. Epub 2023 Mar 10. PMID: 36940641; PMCID: PMC9999249.
-
Essaidi-Laziosi M, Pérez-Rodríguez FJ, Alvarez C, Sattonnet-Roche P, Torriani G, Bekliz M, Adea K, Lenk M, Suliman T, Preiser W, Müller MA, Drosten C, Kaiser L, Eckerle I. Distinct phenotype of SARS-CoV-2 Omicron BA.1 in human primary cells but no increased host range in cell lines of putative mammalian reservoir species. Virus Res. 2024 Jan 2;339:199255. doi: 10.1016/j.virusres.2023.199255. Epub 2023 Nov 6. PMID: 38389324; PMCID: PMC10652112.
R1: For antivirals, a well-known parameter associated with drug activity is protein binding as only the free drug is available to interact with the drug target. While the authors assessed aqueous solubility, they did not try to evaluate protein binding. This is a significant oversight and should be corrected as it has tremendous bearing on the potential clinical utility of these compounds.
Answer: The binding of compounds to plasma proteins represents a pivotal factor in determining the fraction of the drug available for distribution and action within the body. Accordingly, the predictive data obtained by the Deep-PK Predictions software indicate that compounds 4a and 4b have unbound fractions of 42.41% and 36.65%, respectively. This suggests that a substantial proportion of these drugs would be available in free form for distribution and pharmacological action. This information is corroborated by the reliability of the software, which was validated using reference compounds (atazanavir and nirmatrelvir). The free fractions predicted by the software for these compounds were found to be consistent with the experimental data reported in the literature. Therefore, compounds 4a and 4b have the potential for a favorable pharmacokinetic profile, with availability in the systemic circulation and clinical efficacy. This highlights the relevance of their free fractions in the context of plasma protein binding and therapeutic action. We included this information at “2.3.2.2. Physicochemical properties analysis”, “3.3.2. In silico prediction of drug-likeness, physicochemical properties, and pharmacokinetic profile” and “Table S1” in supporting information.
Specific Comments
R1: Line 448: The authors suggest that compounds 4d and 4i are poor inhibitors of Mpro and PLpro which may be the result of a different mechanism of action; however, they did not follow-up with any additional studies. Given the close, structural relationship between all of these compounds, is it possible that the compounds that did show potent inhibition of these proteases could also have more than one mechanism? There should be some additional comment in the discussion section about this finding. In addition, the end of the sentence on lines 448 to 450 (shown below) doesn’t make sense. What is meant by “…indicating the strong interaction and inhibition of the Mpro.”?
“Interestingly, compounds 4d and 4i poorly inhibited both the Mpro and PLpro, suggesting a different mechanism of action, indicating the strong interaction and inhibition of the Mpro.”
Answer: We agree with this referee, and the highlighted text above has been revised in the new manuscript version.
R1: Table 5 and Figure 8: The authors estimate the half-life of 4a to be ~37 hours. This does not appear to be correct. Based on the curve shown in Figure 8, I would estimate the half-life to be <10 hours.
Answer: Since initial phase PK corresponds to an equilibrium between absorption, distribution and clearance, elimination kinetics parameters were estimated at terminal phase, i.e., using last timepoints in the curve. The estimated terminal volume of distribution (Vz/F) and terminal half-life (t½,z) are described as such in both table legend and text.
R1: Line 569: As discussed above, the key driver of clinical utility is not the Mpro IC50 but rather the EC50 for the virus. Given that the Cmax observed following the single IP injection was approximately the same as the EC50 (0.77 uM versus 0.75 uM, respectively), I do not believe the statement of “..suggesting prolonged antiviral activity even after a single i.p. injection.” is accurate and should be removed.
Answer: We agree with this referee, as suggested, the statement was removed from text.
R1: Lines 573-575: As discussed above, the PK parameters for molnupiravir and nirmatrelvir are following oral dosing while the parameters for 4a are following IP dosing. As they are not directly comparable, this section should either be modified with the appropriate disclaimers or deleted.
Answer: We agree with this referee, as suggested, the statement was removed from text

Reviewer 2 Report

Comments and Suggestions for Authors

The manuscript presents a thorough investigation into the design, synthesis, antiviral, and pharmacokinetic properties of a small series of benzocarbazolideneones as Mpro inhibitors, targeting SARS-CoV-2. The study is well-supported by an in silico analysis, providing insights into the interaction between the most active compound and the Mpro viral enzyme.

The authors have effectively articulated the rationale behind their design strategy, and the description of the synthesis of the series is accurate and well-executed. The reported antiviral activity and toxicity data convincingly demonstrate the efficacy of these compounds as SARS-CoV-2 inhibitors. Additionally, enzymatic assays identify MPro as a promising target for compounds 4a and 4b.

The in silico docking results align well with the biological data, and the corresponding descriptions are clear and comprehensive. The in vivo pharmacokinetic evaluation of compound 4a suggests that this compound holds significant potential as an anti-SARS-CoV-2 agent, making it a strong candidate for further investigation.

The authors are to be commended for their thorough work and clear presentation of the results.

However, a few points require correction in Scheme 1:

1.       In Step i, the reaction conditions are incomplete—benzylamine, an important reagent, is missing from the description.

2. In the third step, the manuscript indicates the use of phenyl halides (8) rather than heteroaromatics. Therefore, the final product (9) should reflect the presence of a substituted phenyl ring. Additionally, the inclusion of heteroaromatic derivatives in Scheme 1 should be reconsidered, particularly since the authors mention in Line 383 that the reaction did not proceed in the presence of pyridine (9j). To avoid confusion, heteroaromatic derivatives should be omitted from the scheme.

Author Response

Referee 2
The manuscript presents a thorough investigation into the design, synthesis, antiviral, and pharmacokinetic properties of a small series of benzocarbazolideneones as Mpro inhibitors, targeting SARS-CoV-2. The study is well-supported by an in silico analysis, providing insights into the interaction between the most active compound and the Mpro viral enzyme.
The authors have effectively articulated the rationale behind their design strategy, and the description of the synthesis of the series is accurate and well-executed. The reported antiviral activity and toxicity data convincingly demonstrate the efficacy of these compounds as SARS-CoV-2 inhibitors. Additionally, enzymatic assays identify MPro as a promising target for compounds 4a and 4b.
The in silico docking results align well with the biological data, and the corresponding descriptions are clear and comprehensive. The in vivo pharmacokinetic evaluation of compound 4a suggests that this compound holds significant potential as an anti-SARS-CoV-2 agent, making it a strong candidate for further investigation.
The authors are to be commended for their thorough work and clear presentation of the results.
However, a few points require correction in Scheme 1:
1. In Step i, the reaction conditions are incomplete—benzylamine, an important reagent, is missing from the description.
2. In the third step, the manuscript indicates the use of phenyl halides (8) rather than heteroaromatics. Therefore, the final product (9) should reflect the presence of a substituted phenyl ring. Additionally, the inclusion of heteroaromatic derivatives in Scheme 1 should be reconsidered, particularly since the authors mention in Line 383 that the reaction did not proceed in the presence of pyridine (9j). To avoid confusion, heteroaromatic derivatives should be omitted from the scheme.
Answer: Thank you very much for your thoughtful feedback! We appreciate your comments. We have made the necessary corrections to Scheme 1 to address the misconceptions you pointed out, ensuring clearer understanding.

Round 2

Reviewer 1 Report

Comments and Suggestions for Authors

The authors have adequately addressed the concerns raised during the initial review of the manuscript.